# Phenotypic Spectrum of *NFIA* Haploinsufficiency: Two Additional Cases and Review of the Literature

**DOI:** 10.3390/genes13122249

**Published:** 2022-11-30

**Authors:** Veronica Bertini, Francesca Cambi, Alessandro Orsini, Alice Bonuccelli, Aureliano Fiorini, Andrea Santangelo, Massimo Scacciati, Maurizio Elia, Ornella Galesi, Diego Peroni, Angelo Valetto

**Affiliations:** 1Cytogenetic Unit, Department of Laboratory Medicine, Azienda Ospedaliero-Univeristaria Pisana, Via Roma 57, 56100 Pisa, Italy; 2Pediatric Neurology, Pediatric Department, Santa Chiara University Hospital, Azienda Ospedaliero-Univeristaria Pisana, Via Roma 57, 56100 Pisa, Italy; 3Oasi Research Institute—IRCCS, 94018 Troina, Italy

**Keywords:** nuclearfactor I family, macrocephaly, NCS defects, corpus callosum agenesis/hypoplasia, bilateral proximally placed first fingers, neurodevelopmental disorders

## Abstract

The *NFIA* (nuclear factor I/A) gene encodes for a transcription factor belonging to the nuclear factor I family and has key roles in various embryonic differentiation pathways. In humans, *NFIA* is the major contributor to the phenotypic traits of “Chromosome 1p32p31 deletion syndrome”. We report on two new cases with deletions involving *NFIA* without any other pathogenic protein-coding gene alterations. A cohort of 24 patients with *NFIA* haploinsufficiency as the sole anomaly was selected by reviewing the literature and public databases in order to analyze all clinical features reported and their relative frequencies. This process was useful because it provided an overall picture of the phenotypic outcome of *NFIA* haploinsufficiency and helped to define a cluster of phenotypic traits that can facilitate clinicians in identifying affected patients. *NFIA* haploinsufficiency can be suspected by a careful observation of the dysmorphisms (macrocephaly, craniofacial, and first-finger anomalies), and this potential diagnosis is strengthened by the presence of intellectual and developmental disabilities or other neurodevelopmental disorders. Further clues of *NFIA* haploinsufficiency can be provided by instrumental tests such as MRI and kidney urinary tract ultrasound and confirmed by genetic testing.

## 1. Introduction

“Chromosome 1p32p31 deletion syndrome”, also referred as “brain malformations with or without urinary tract defects” (BRMUTD) (OMIM # 613735), is a genomic disorder characterized by macrocephaly, CNS malformations, congenital anomalies of the kidneys and the urinary tract, developmental delay (DD), intellectual disability (ID), and facial dysmorphism. Other clinical findings may include cryptorchidism, inguinal hernia, anomalies of external genitalia, cutis marmorata, syringomyelia, congenital heart defects, and Juvenile Moyamoya [1,2,3,4,5,6,7]. The deleted region includes the *NFIA* gene (nuclear factor I/A) (OMIM * 600727; ENSG00000162599), which is responsible for the main phenotypic traits of this syndrome.

*NFIA* is a gene located on chromosome 1p31.3 that encodes for a transcription factor belonging to the nuclear factor I family of dimeric DNA binding proteins, along with *NFIB, NFIC*, and *NFIX* [8]. This protein has three domains: (1) an N-terminal highly conserved region, whose function is not known; (2) a DNA binding/dimerization domain (MH1), and (3) a C-terminal transactivation and/or repression domain (CTF_NFI). The MH1 domain binds to a nucleotide consensus sequence within the promoter region of several genes, whereas the CTF_NFI domain activates basal transcription factors at transcription start sites by displacing repressive histones from target genes and interacting with other coactivator proteins [9]. *NFIA* gives rise to 31 transcript variants, but the functional significance of each isoform has not been yet clarified. The MANE transcript (matched annotation from NCBI and EMBL-EBI) is *NFIA-207* (ENST00000403491.8), whichis encoded by 11 exons, extending from position 61,548,233 to 61,928,460 (GRCh37/hg19), and has ubiquitous expression (https://www.gtexportal.org/, accessed on 20 September 2022).

According to in vitro analyses and studies of animal models, *NFIA* has a key role in various differentiation pathways during embryogenesis. It is a crucial regulator of articular cartilage differentiation [10] and controls the adipogenic and myogenic gene program to ensure brown and beige adipocyte differentiation [11]. *Nfia* knock-out mice show hydrocephalus and agenesis of the corpus callosum, and, at reduced penetrance, exhibit abnormalities of the ureteropelvic and ureterovesical junctions, as well as bifid and megaureters [1,8]. NFI proteins play an important role in the development of the central nervous system (CNS), including axon guidance with outgrowth and in glial or neuronal cell differentiation and migration [8,12,13,14,15].

In this paper, we focused on cases with *NFIA* haploinsufficiency as the sole anomaly, excluding all those patients with other pathogenic protein-coding gene alterations. We report on two new individuals and reviewed the literature and public databases to better delineate the phenotypic spectrum of *NFIA* haploinsufficiency and its contribution to “Chromosome 1p32p31 deletion syndrome”.

## 2. Materials and Methods

### 2.1. Molecular Analysis

Array CGH was performed. Genomic DNA of the patients was isolated from peripheral blood by standard methods. DNA from healthy subjects was used as controls. Test and reference DNA were differentially labeled with Cy5-dCTP or with Cy3-dCTP using random primer labeling and applied to 60K arrays, according to the manufacturer’s protocol (case 1: Agilent, Santa Clara, CA, USA; case 2: Technogenetics, Milan, Italy). For each case, the array-CGH was repeated twice in order to confirm the results. Since quality of the experiment influences the Copy Number Variants (CNVs) (i.e., number and type of calls and probes included/excluded in each call), we elaborated only experiments that met the ‘excellent’ criteria as determined by the QC report. The CNV analysis was performed according to the guidelines of the Italian Society of Human Genetics (https://www.sigu.net, accessed on 20 September 2022) and to the American College of Medical Genetics guidelines [16].

### 2.2. Bioinformatic Analyses

CNVs classification was performed using the Database of Genomic Variants (DGV) (http://projects.tcag.ca/variation, accessed on 20 September 2022) and the University of California Santa Cruz (UCSC) Genome Browser (https://genome.ucsc.edu, accessed on 20 September 2022).PubMed and Online Mendelian Inheritance in Man (OMIM) were also consulted for evaluating genotype–phenotype association.

Our search for patients with *NFIA* haploinsufficiency, including CNVs and SNVs (single nucleotide variants), was performed in the DECIPHER (https://www.deciphergenomics.org/, accessed on 20 September 2022), ClinVar, Leiden Open Variation, and HGMD databases.

*NFIA* transcripts were identified using Ensembl. Data on the expression profiles were assessed using GTEx (https://www.gtexportal.org/, accessed on 20 September 2022) and the UCSC Genome Browser.

### 2.3. Patients

#### 2.3.1. Patient 1

Patient 1 (P1) is a 2-year-old male toddler who was referred to the pediatric unit at the age of 10 months for macrocephaly.

At time of referral, his height was 81 cm (>98th centile), his weight was 12.6 kg (>98th centile), his head circumference was 53 cm (>97th centile), and his BMI was 19.2 (85–97th centile). The patient was born at 37 weeks of gestation with a normal delivery following a pregnancy complicated by cholecystitis at 34 weeks of gestation. His birth weight was 3390g (90th centile), length 49 cm (64th centile), head circumference was 38 cm (>100th centile). Family history was unremarkable.

On physical examination he presented with macrocephaly, scaphocephaly, a high and wide forehead with evident frontal bossing, and low-set ears. He showed brachydactyly with bilateral proximally placed first fingers and short lower limbs. The eye examination revealed the presence of a mild strabismus, whereas hearing screening, blood tests, and metabolic and hormonal work up were all normal. EEG revealed minor, nonspecific discharges in the fronto-central-parietal regions. No epileptic seizures have been recorded. A cranial ultrasound, performed at the age of 3 months, revealed sagittal synostosis, whereas the metopic, lambdoid, and coronal sutures were still open. A brain MRI showed diffuse corpus callosum hypoplasia and a dysmorphic aspect of the cerebral ventriculus with widening of the anterior portions of the horns. No Chiari I malformation was detected.

At the first evaluation, the patient showed normal psychomotor development, but, at follow-up (age 2 years), he presented mild/moderate DD with severe delay in language (absence of speech) and hyperactivity disorder. An appearance of congestion of the optic disc was reported in anocular ultrasound, but the ophthalmological examination was negative. A brain MRI showed no alteration of the cranial nerves but confirmed the dysmorphic aspects already described previously; in particular, it confirmed the hypoplasia of the cerebral commissural system (corpus callosum and anterior commissure) with dysmorphic appearance of the lateral ventricles. An EEG showed focal abnormalities in posterior-temporal regions, with more in the right hemisphere. However, the patient did not present with epileptic seizures.

#### 2.3.2. Patient 2

Patient 2 (P2) is a 22-year-old womanfollowed by an outpatient basis for ID (IQ 62), oppositional defiant disorder (ODD), and trichotillomania. Physical examination showed a height of 158 cm (25th centile), weight of 47 kg (10th centile), and a normal head circumference at the upper limits (56.5 cm, 75th centile).

Slight facial dysmorphisms were shown, such as a high forehead, downslanting palpebral fissures, low-set and posteriorly rotated ears, a small and poorly structured philtrum, an open bite, hypotrophy of thenar and hypothenar eminences, and bilateral proximally placed first fingers. No skin dyschromia was revealed. An MRI showed a thin corpus callosum, Chiari malformation type I, hypoplastic rectus sinus, and a vicariant vessel along the right profile of the trunk. Neither ventricles anomies nor craniosynostosis were detected. Heart, lung, and abdomen ultrasound was normal. No epilepsy or EEG anomalies were reported. The patient was no longer available for follow up.

## 3. Results

Array CGH detected in P1 a de novo 1.484 kb deletion, starting from position 60,568,797 and ending at position 62,052,980 (GRCh37/hg19). The deleted region includes *LINC01748*, *LOC101926964*, *NFIA-AS2*, *NFIA-AS1,* and *NFIA*.

P2 showed a 27 kb deletion, starting from position 61,818,169 and ending at position 61,845,524 (GRCh37/hg19). The deleted region includes part of exon 5 and the entirety of exon 6 of *NFIA* (ENST00000403491.8; RefSeq ID NM_001134673). A segregation pattern could not beestablished.

These results are reported in Table 1, along with the genetic alterations of the selected patients [17,18,19,20,21,22,23,24,25].

The pathological traits of these patients have been grouped into macroareas: dysmorphisms (Table 2), neurological/behavioral abnormalities, renal/urinary trait alterations (Table 3), and CNS malformations (Table 4). In each table are listed only those patients where the clinical traits were evaluated. The few additional defects not included in these categories are reported in Appendix A.

## 4. Discussion

Here, we report on two new individuals carrying microdeletions solely affecting *NFIA* without any other flanking protein-coding sequences. In order to further define the phenotypic effects of *NFIA* haploinsufficiency, we reviewed the literature and public databases excluding those patients without detailed clinical information or individuals with pathogenic deletions, translocations, inversions, or genetic variants disrupting additional protein-coding sequences [1,2,3,4,5,6,7,23,26,27,28,29] (Appendix A). This careful selection allowed for the identification of a cohort of 24 patients with pathogenic alterations solely involving *NFIA*.

In Table 1 are listed our two cases along with the other 24 selected patients; their age ranges from 1- to 42-years-old; the prevalence of males and females is comparable (12 males versus 14 females); in four cases, the variants have been inherited, and twelve were de novo [17,18,19,20,21,22,23,24,25].

The precise localization of *NFIA* pathogenic variants is also reported. Since alteration of this gene hasoften been referred to alternative transcripts, we have converted each of them according to the *NFIA-207* (ENST00000403491.8; RefSeq ID NM_001134673) (Appendix A). As shown in Figure 1, SNVs do not have one hotspot and are spread over various functional domains; some microdeletions are intragenic and involve only few exons, while others disrupt all of the coding sequence with theirregulatory elements (*NFIA-AS2*, *NFIA-AS1*) and further noncoding RNAs (*LINC01748*, *LOC101926964*).

To better evaluate the phenotypic outcome of *NFIA* haploinsufficiency, the pathological traits of the selected patients have been grouped into macroareas: dysmorphisms (Table 2), neurological/behavioral abnormalities, renal/urinary trait alterations (Table 3), and CNS malformations (Table 4). In each table are listed only those patients where the clinical traits were evaluated. The few additional defects not included in these categories are reported in Appendix A and CNS malformations (Table 4).

The major clinical characteristics are summarized in Table 5 along with their frequency. The prevalence of each single trait was calculated both by the ratio between the presence of a clinical sign on the number of patients where this clinical sign has been evaluated and the ratio of all the 26 selected patients.

***Dysmorphisms*** (Table 2). Among physical anomalies, macrocephaly is a trait almost invariably present (except P2); high forehead and a low-set ears are other recurrent signs, often associated with nonspecific facial-dysmorphic features.

Craniofacial anomalies, including asymmetries and craniosynostosis, represent a distinctive trait described in most patients. Metopic, lambdoid, or sagittal craniosynostosis were not observed in four cases (P4, P5, P6, P8). Three of them are members of the same family; thus, it was hypothesized that these anomalies could be the consequence of additional familiar genetic modifiers; our report of an additional case with craniosynostosis (P1) strengthens this correlation with haploinsufficiency. It is worth noting that in chicken, *NFIA* participates in osteoblast differentiation by interacting with the Wnt and Ihh pathways [10].

Bilateral proximally placed first fingers seem to be another characteristic dysmorphism, present in our patients (P1, P2) and in four other cases (P6, P8, P11, P12). This sign may have been overlooked, and it is possible that its frequency is underestimated.

Macrocephaly, craniofacial abnormalities, and first-finger anomalies are quite peculiar traits, and their concurrence can represent a first diagnostic clue for *NFIA* haploinsufficiency.

***Neurological/behavioral abnormalities*** (Table 3). DD and/or ID are invariably present in this disorder, ranging from very mild to severe (details are available in Appendix A). Beside ID and DD, this cohort shows the concurrence of other NDDs, including hyperactivity disorder (P1), ODD (P2), ADHD-Combined Type (P4), PDDNOS (P4), as well as neurological signs such as facial asymmetry (P9), mild spasticityat lower limbs (P9), extremely low PSI (P16), and epilepsy (P16, P17, P18, P21).It is worth noting that the comorbidity is an almost constant feature of NDDs because different NDDs can represent the variable expressivity of the same genetic alteration [30]. The comorbidity of this cohort strengthens this hypothesis. Hypotonia, exotropia, and hearing impairment are other neurological signs reported.

***Renal/urinary trait and CNS defects*** (Table 3 and Table 4). Renal/urinary tract and CNS defects are considered to bethe key features of “Chromosome 1p32-p31 deletion syndrome”, whichis also referred to as “brain malformations with or without urinary tract defects” (BRMUTD) (OMIM # 613735).

In cases with *NFIA* haploinsufficiency, the frequency of renal/urinary tract abnormalities is probably lower than previously supposed [1], and in this cohort, they were excluded in twelve cases (Table 3). The defects described are heterogeneous, even when intrafamilial (i.e., P5 versus P7). Hydronephrosis and renal cysts are the most common clinical signs, even if each of them is reported only in three patients (P4, P7, P15 and P5, P9, P17, respectively). The absence of recurrent clinical features, along with their low frequency, make the renal/urinary tract anomalies not very helpful for the identification of this disorder. As experiments in knocked-out mice highlighted the involvement of *NFIA* in the embryonic development of the excretory apparatus [1], a surveillance of kidneys and urinary trait is mandatory when haploinsufficiency of this gene is detected.

On the other hand, CNS alterations represent a key trait of *NFIA* haploinsufficiency. Corpus callosum and cerebral ventricles anomalies are the most distinctive signs: the corpus callosum can be thin, hypoplastic, absent, or dysgenetic, and the presence of ventriculomegaly or hydrocephalus is quite constant. Along with these defects, a constellation of less common and heterogeneous CNS anomalies has been described, such as cortical malformations (P11, P12, P15, P21, P26), hypoplasia or atrophy of brain areas (P5, P11, P12, P19), decreased periventricular white matter (P3, P11, P25), partial incomplete inversion of the left hippocampi (P5, P11), and cyst presence (P11, P15, P25). P2 shows the presence of cerebral vascular malformations that have never been reported before. All these anomalies confirm the key role of *NFIA* in the differentiation of multiple brain areas during embryonic development [12,13,14,15].

From Table 1, Table 2, Table 3 and Table 4, a strict correlation between the genetic variants and clinical signs does not emerge: greater deletions do not correspond to a more severe phenotype; moreover, familialcases that share the same anomaly show a clinical spectrum, confirming a variable expressivity of some phenotypic traits.

## 5. Conclusions

*NFIA* is a gene with a key role in various differentiation pathways during embryogenesis; therefore, its haploinsufficiency is thought to determine a pleiotropic effect.

In this article, two new cases with deletions solely involving *NFIA* are reported, and the literature and public databases were reviewed in order to evaluate the phenotypic outcome of *NFIA* haploinsufficiency. First, all those patients with *NFIA* alterations and pathogenic genetic variants disrupting additional protein-coding sequences were excluded, since a thoughtful selection of cases is the most important step to accurately delineate genotype–phenotype correlations of a genetic disorder. Second, all the clinical signs and their relative frequencies in the 24 selected cases were analyzed. This process was useful to have an overall picture of phenotypic outcome of *NFIA* haploinsufficiency and to define a cluster of phenotypic traits that can facilitate clinicians in identifying the affected patients (Table 5). The potential diagnosis of *NFIA* haploinsufficiency can be advanced by a careful observation of the dysmorphisms (macrocephaly, craniofacial, and first-finger anomalies) and be strengthened by the presence of ID/DD or other NDDs. Instrumental tests such as MRI and kidney urinary tract ultrasound can provide key elements to further strengthen this hypothesis, which can be confirmed by genetic testing. Additional cases with well-described phenotypes will be helpful to better elucidate the full clinical spectrum.

The present analysis confirms that *NFIA* is responsible for the main phenotypic traits of Chromosome 1p32p31 deletion syndrome, even if other genes can contribute to modify the severity of some signs, such as ID/DD [6]. Less common traits of this syndrome, such as cryptorchidism, inguinal hernia, anomalies of external genitalia, syringomyelia, congenital heart defects, and Juvenile Moyamoya [1,2,3,4,5,6,7], were never reported in this selected cohort with only *NFIA* haploinsufficiency, and they are probably caused by dosage alterations of different deleted genes.

## Figures and Tables

**Figure 1 genes-13-02249-f001:**
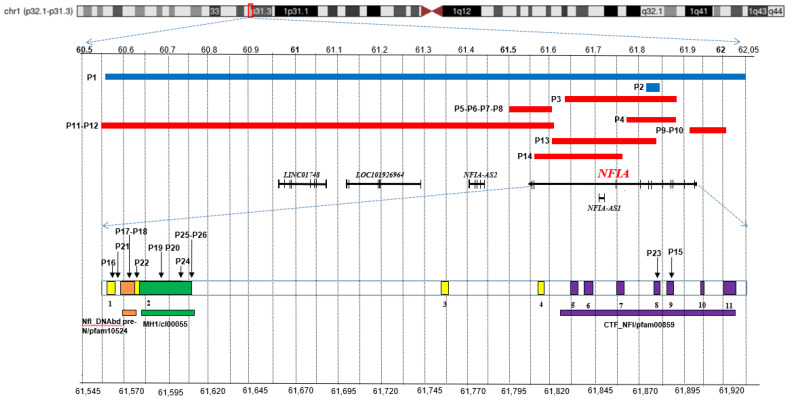
**Overview of *NFIA* pathogenic variants.** Chromosome 1 ideogram with highlight on p32.1-p31.3 region. ***Top***: the blue bars indicate the deletions detected in the patients here reported (P1-P2), and the red ones correspond to cases described in literature and public databases (P3-P14). ***Middle***: coding sequences, including NFIA-207 (ENST00000403491.8; RefSeq ID NM_001134673) and other non-coding RNAs, are represented by dark bars. ***Bottom***: a NFIA-207 zoom in that highlights the exons (1-11) along with the SNVs of patients P15-P26. The conserved domains are also depicted: NfI_DNAbd_pre-N/pfam10524 (orange), MH1/cI00055 (green), CTF_NFI/pfam00859 (purple). The top and bottom scales refer to GRCh37/hg19. Relative exons lengths are not to scale.

**Table 1 genes-13-02249-t001:** Patients with solely *NFIA* haploinsufficiency.

DELETIONS						
Reference	Subjects	Sex/Age	Position (GRCh37/hg19)	Extent (kb)	NCBI RefSeq Genes (UCSC) *NFIA-207*	Inheritance
In this report	P1	M/2 year	1:60,568,797–62,052,980	1.484	LINC01748, LOC101926964, *NFIA-AS2, **NFIA*** (ex 1–11), *NFIA-AS1*	dn
In this report	P2	F/22 year	1:61,818,169–61,845,524	27	***NFIA*** (ex 5–6)	NR
Mikhail et al., 2011 [17]; Hollenbeck et al., 2016 [18] (Patient 99199)	P3	F/25 year	1:61,632,666–61,886,758	254	***NFIA*** (ex 3–9) *NFIA-AS1*	not mat NR pat
Rao et al., 2014 [19]	P4	F/8 year	NR	120	***NFIA*** (ex 4–9)	dn
Nyboe et al., 2015 [20]	P5 (I-1) **P6** (II-1) P7 (II-2) P8 (II-3)	P5: M/42 year P6: F/13 year P7: M/10 year P8: M/6 year	1:61,497,698–61,607,171	109	***NFIA*** (ex 1–2)	P5: NR P6-P7-P8: pat
Hollenbeck et al., 2016 [18] (Patient 13857)	**P9** P10 (father)	P9: M/1.5 year P10: M/NR	NR	99	***NFIA*** (ex 11)	P9: pat P10: NR
Bayat et al., 2017 [21]	**P11** P12 (mother)	P11: F/9 year P12: M/37 year	1:60,549,342–61,614,478	1.065	LINC01748, LOC101926964 *NFIA-AS2, **NFIA*** (ex 1–2)	P11: mat P12: NR
288170 DECIPHER	P13	F/NR	1:61,616,698–61,845,603	229	*NFIA-AS1, **NFIA*** (ex 3–6)	dn
358646 DECIPHER	P14	F/1 year	1:61,557,457–61,770,053	213	*NFIA-AS1, **NFIA*** (ex 3)	dn
**SNVs**						
	**Subjects**	**Sex/Age**	**NM_001134673**	**NP_001128145.1**	**Type of Mutation**	**Inheritance**
Negishi et al., 2015 [22]	P15	M/5 year	c.1094delC	p.Pro365-HisfsTer32	stop_gained/pathogenic	dn
Revah-Politi et al., 2017 [23] (patient 1)	P16	F/18 year	c.25_26dupCC	p.Gln9ProfsTer49	stop_gained/likely pathogenic	dn
Revah-Politi et al., 2017 [23] (patient 2), (patient 3)	**P17** P18	P17: F/7 year P18: F/35 year	c.70C > T	p.Arg24Ter	stop_gained/pathogenic	P17: matP18: NR
Zhang et al., 2020 [24]	P19	M/3 month	c.220C > T	p.Arg74Ter	stop_gained/pathogenic	dn
264161 DECIPHER	P20	F/2 year 9 month	c.224T > C	p.Leu75Pro	missense/likely pathogenic	dn
282711 DECIPHER	P21	F/13 year	c.28-2A > G	SpliceAI: ΔS acceptor loss	splice_acceptor/pathogenic	dn
291368 DECIPHER	P22	M/2 year 9 month	c.112C > T	p.Arg38Ter	stop_gained/likely pathogenic	dn
305439 DECIPHER	P23	M/1 year	c.1051C > T	p.Arg351Ter	stop_gained/pathogenic	NR
435805 DECIPHER	P24	F/13 year	c.500A > G	p.His167Arg	missense/ likely pathogenic	mat
Uehara et al., 2021 [25] (patient 1)	P25	F/6 year	c.373A > G	p.Lys125Glu	missense/ likely pathogenic	dn
Uehara et al., 2021 [25] (patient 2)	P26	M/14 month	c.373A > G	p.Lys125Glu	missense/ likely pathogenic	dn

For each patient, the corresponding references (literature and databases), age, sex, and inheritance are reported. The deletions are described with the position of the first and last abnormal probe (GRCh37/hg19), the extent, the exons of *NFIA-207* (ENST00000403491.8; RefSeq ID NM_001134673), and other non-protein-coding sequences included (top). The single nucleotide variants (SNVs) are reported according to NM_001134673 with the corresponding amino acid changes and type of mutation (bottom). SNVs classification was performed using Varsome (https:varsome.com/, accessed on 20 September 2022).dn: de novo; ex: exon; F: female; M: male; mat: maternal; NR: not reported; P: patient; pat: paternal. In familiar cases, the proband is in bold.

**Table 2 genes-13-02249-t002:** Dysmorphisms.

Patient	Macrocephaly	High Forehead	Low- Set Ears	Facial Dysmorphic Features	CraniofacialAnomalies	Hands and Feet Anomalies		Other Anomalies
						Bilateral Proximally Placed First Fingers	Other	
P1	Y	Y	Y	frontal bossingmild strabismus	sagittal synostosis scaphocephaly	Y	brachydactyly	short lower limbs
P2	N	Y	Y	downslanting palpebral fissures, posteriorly rotated auricles, small and poorly structured philtrum, open bite	NO craniosynostosis	Y	hypotrophy of thenar and hypothenar eminences	-
P3 [17,18]	Y	Y	Y (left)	high palate, pointed chin hypotelorism	-	-	-	scarce hair, scoliosis, webbed neck
P4 [19]	Y	Y *	Y *	upslanting palpebral fissures, broad anteverted nose, overfolded helices	metopic synostosis	-	-	cutis marmorata
P5 [20]	Y	-	-	-	sagittal synostosis	-	-	overgrowth
P6 [20]	Y	Y *	Y	downslanting palpebral fissures	lambdoid synostosis	Y	-	overgrowth
P7 [20]	Y	-	Y	-	-	-	-	overgrowth
P8 [20]	Y	-	-	-	lambdoid synostosis	Y	-	overgrowth
P9 [18]	-	-	-	right eye exotropia, mild ptosis	-	-	-	-
P11 [21]	Y	Y *	Y	downslanting palpebral fissures	craniofacial asymmetry	Y	bilateral slightly broad first fingers	-
P12 [21]	Y	Y *	Y	high palate	craniofacial asymmetry	Y	-	-
P13	Y	-	-	-	-	-	-	-
P14	Y	-	-	-	-	-	-	-
P15 [22]	Y	Y	-	-	-	-	-	-
P16 [23]	Y	-	-	-	-	-	small hands and feet	obesity
P17 [23]	Y	Y	-	frontal bossing	-	-	-	bilateral radioulnar synostoses
P18 [23]	Y	-	-		-	-	-	-
P19 [24]	Y	Y *	Y *	hypertelorism, slightly pointed chin, large ears	-	-	-	-
P20	-	-	-	-	abnormal facial shape	-	-	-
P21	Y	Y	-	short nose, wide nasal bridge	-	-	-	melanocytic nevus, papule, supernumerary nipple
P22	Y	-	-	-	abnormal facial shape	-	-	genu valgum, pes planus
P23	Y	-	-	-	-	-	broad hallux, broad thumb, short distal phalanx of the thumb	-
P24	Y	-	-	-	-	-	-	-
P25 [25]	Y	Y *	-	small eyes, anteverted nares, depressed nasal bridge, a broad columella, a thin upperlip, high arched palate, exotropia	-	-	-	-
P26 [25]	Y	Y	-	thick eyebrow, short nose, anteverted nares, long philtrum, thin upperlip, retrognathia	-	-	-	-

For each patient, the corresponding references are reported. The shared clinical signs are highlighted in the same color. -: not evaluated; N: absent; Y: present; *: deduced by the picture.

**Table 3 genes-13-02249-t003:** Neurological/behavioral abnormalities and renal/urinary tract defects.

Neurological/Behavioral Abnormalities						Renal/Urinary Tract Defects
Patient	ID	DD	Seizure	NDDs	Other	
P1	Y	Y	N	hyperactivity disorder, language and speech delay		N
P2	Y	-	N	ODD	trichotillomania	N
P3 [17,18]	Y	-	-	-	bipolar disorder/depression, incapability of making her own decisions	N
P4 [19]	Y	Y	-	PDDNOS, ADHD-Combined Type	hypotonia	hydronephrosis, renal calculus, kinking of the pelvic–ureteric junction
P5 [20]	N	N	-	-	-	two renal cysts
P6 [20]	Y	Y	-	-	-	N
P7 [20]	Y	Y	-	-	-	right hydronephrosis and hydrourethra, small ureterocele, frequent urinary tract infections
P8 [20]	Y	Y	-	-	-	N
P9 [18]	-	Y		mild lower-extremity spasticity, asymmetric movement of facial muscles		renal cysts
P11 [21]	Y	Y	-	-	-	-
P12 [21]	Y	Y	-	-	-	N
P13	Y	-	-	-	-	N
P14	Y	-	-	-	-	-
P15 [22]	-	Y	-	-	-	-
P16 [23]	Y	Y	-	-	-	left hydronephrosis, cystectasia, bilateral grade IV vesicoureteral reflux
P17 [23]	N	Y	Y	extremely low PSI	hypotonia	N
P18 [23]	-	Y	Y	-	bilateral hearing loss, photophobia headaches, nonspecific complaints of arm and leg pain	renal cyst, urinary retention, frequent urinary tract infections
P19 [24]	N	N	Y	-	headaches and/or migraines, depression	N
P20	Y	Y	-	-	impaired left ear auditory brainstem response, bilateral ametropia	N
P21	Y	-	-	-	sleep disturbance	nephrolithiasis
P22	-	Y	Y	-	hypotonia	-
P23	-	Y	-	-	-	-
P24	-	Y	-	-	-	-
P25 [25]	Y	Y	-	-	congenital hearing loss	N
P26 [25]	-	Y	-	-	mild congenital hearing impairment	N

For each patient the corresponding reference is reported. The shared clinical signs are highlighted in the same color. -: not evaluated; N: absent; Y: present; ADHD-Combined Type: Attention Deficit Hyperactivity Disorder Combined Type; DD: developmental delay; ID: intellectual disabilities; ODD: Oppositional Defiant Disorder; NDDs: neurodevelopmental disorders; PDDNOS: Pervasive Developmental Disorder—Not Otherwise Specified; PSI: processing speed.

**Table 4 genes-13-02249-t004:** CNS defects.

Patient	Corpus Callosum Anomalies	Ventricular Anomalies	Other
P1	hypoplasia	dysmorphic aspect of the cerebral ventriculus with widening of the anterior portions of the horns	no Chiari I malformation
P2	thin	N	Chiari I malformation, cerebral vascular malformations
P3 [17,18]	hypoplasia	mild hydrocephalus	diffusely decreased volume of white matter, mild tonsillar ectopia, no Chiari I malformation
P4 [19]	hypoplasia	ventriculomegaly, partial fusion of lateral ventricles	partial absence of mid/posterior septum pellucidum
P5 [20]	hypoplasia	N	absent falx cerebri, partial incomplete inversion of the left hippocampi
P6 [20]	hypoplasia	ventriculomegaly	-
P8 [20]	hypoplasia	ventriculomegaly	herniation of cerebellar tonsils
P9 [18]	-	-	prominent cavum septum pellucidum and cavum vergae, tethered spinal cord
P11 [21]	thin	ventriculomegaly with asymmetric widened, edged lateral ventricles, widened third ventricle	loss of/missing white matter, frontalcortical malformation with polymicrogyria, hypoplastic falx cerebri with interdigitated frontal gyri, bilateral partial incomplete inversion of the hippocampi, arachnoid cysts in the posterior fossa
P12 [21]	hypoplasia	mild ventriculomegaly	mild frontoparietal atrophy/hypoplasia, slight asymmetry of frontal gyri
P14	agenesis	-	-
P15 [22]	agenesis	ventricular enlargement	interhemispheric cysts, polymicrogyria in the right frontal lobe
P16 [23]	agenesis of the rostral part	ventriculomegaly	Chiari I malformation
P17 [23]	dysgenesis	hydrocephalus	-
P18 [23]	N	N	Chiari I malformation
P19 [24]	thin	enlarged bilateral cerebral ventricles	bilateral frontal and temporal brain atrophy
P21	hypoplasia	ventriculomegaly	abnormality of neuronal migration
P23		ventriculomegaly	-
P25 [25]	thin	ventricular enlargement, ventricular wall irregularity	cyst of septipellucidi, decreased white matter volume
P26 [25]	hypoplasia	-	polycerebral gyrus at parasylvius fissures, cortical dysplasia of bilateral cerebral hemisphere, partial myelination delay

For each patient, the corresponding reference are reported. The shared clinical signs are highlighted in the same color. -: not evaluated; N: absent.

**Table 5 genes-13-02249-t005:** Major clinical features.

CLINICAL FEATURES	A	B
**DYSMORPHISMS**		
Macrocephaly	22/23 (96%)	22/26 (85%)
High forehead	13/13 (100%)	13/26 (50%)
Low-set ears	9/9 (100%)	9/26 (35%)
Facial dysmorphic features	13/13 (100%)	13/26 (50%)
Craniofacial anomalies	9/10 (90%)	9/26 (35%)
Bilateral proximally placed first fingers	6/6 (100%)	6/26 (26%)
Other hands and feet anomalies	5/5 (100%)	5/26 (19%)
Other	11/11 (100%)	11/26 (42%)
**NEUROLOGICAL/BEHAVIORAL ABNORMALITIES**		
ID	15/18 (83%)	15/26 (58%)
DD	18/20 (90%)	18/26 (69%)
Seizure	4/6 (67%)	4/26 (15%)
NDDs	5/5 (100%)	5/26 (19%)
Other	11/11 (100%)	11/26 (42%)
**RENAL/URINARY TRACT DEFECTS**		
Hydronephrosis	3/19 (16%)	3/26 (11%)
Renal cysts	3/19 (16%)	3/26 (11%)
Other	5/19 (26%)	5/26 (19%)
**CNS DEFECTS**		
Corpus callosum anomalies	17/18 (94%)	17/26 (65%)
Ventricular anomalies	14/17 (82%)	14/26 (54%)
Other	16/16 100%)	16/26 (61%)

Frequency of the main clinical features. The prevalence of each single trait is calculated both by the ratio between the presence of this clinical sign on the number of patients where this clinical sign has been evaluated (column A) and the ratio of all the selected patients (column B).

## Data Availability

Publicly available datasets were analyzed in this study. Thesedata can be found at: https://www.deciphergenomics.org/ (accessed on 20 September 2022); https://www.genome.ucsc.edu/ (accessed on 5 April 2022); https://omim.org/ (accessed on 20 September 2022); https://www.ensembl.org (accessed on 20 September 2022); https://www.gtexportal.org/home (accessed on 20 September 2022); http://projects.tcag.ca/variation (accessed on 20 September 2022); http://www.ncbi.nlm.nih.gov/pubmed (accessed on 20 September 2022); https://www.sigu.net (accessed on 20 September 2022);ClinVar (https://www.ncbi.nlm.nih.gov/clinvar, accessed on 20 September 2022); Leiden Open Variation Database (https://www.lovd.nl, accessed on 20 September 2022); the Human Gene Mutation Database (https://www.hgmd.cf.ac.uk/ac/index.php, accessed on 20 September 2022); and Varsome (https:varsome.com, accessed on 16 November 2022).

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
