# Peer review of "Phenotypic Spectrum of NFIA Haploinsufficiency: Two Additional Cases and Review of the Literature"

_genes, 2022, doi:10.3390/genes13122249_

Round 1

Reviewer 1 Report

Bertini and colleagues describe the phenotypic spectrum of NFIA haploinsufficiency. I have attached a document with changes. I also have several questions/concerns. 

First, a table (or combined into the current tables) or figure stating the over prevalence of each phenotype is necessary. As of now, the paper just reads as a list of phenotypes, not a cohesive phenotypic spectrum of NFIA haploinsufficiency. I also think that the tables should have more of the shared phenotypes as columns. The tables should be in the results, not discussion.

Currently, this paper is relevant for a small audience interested in NFIA, but I think even that audience will be confused by some of the things in this paper, which I have pointed out in the attachment (which has some formatting issues, I apologize). I think review of similar types of manuscripts would be useful in finding the best ways to present this data. I also wonder if there are any statistical analyses that could be done (maybe compare to NFIX disorder?)

You claim to focus on haploinsufficiency, but also have missense variants in your table. Is there evidence that these impair protein function? You could have a table with in silico predictions.

You do not need to have as many links in the text as you do, as many of these resources have papers that can be cited.

In Figure 1, a protein level map would be nice to see if they cluster in any domains.

Author Response

We thank the Reviewer for her/his careful reading of the manuscript and the insightful suggestions

Bertini and colleagues describe the phenotypic spectrum of NFIA haploinsufficiency. I have attached a document with changes. I also have several questions/concerns.

First, a table (or combined into the current tables) or figure stating the over prevalence of each phenotype is necessary. As of now, the paper just reads as a list of phenotypes, not a cohesive phenotypic spectrum of NFIA haploinsufficiency. I also think that the tables should have more of the shared phenotypes as columns. The tables should be in the results, not discussion.

We thank for the suggestion, we have moved tables 2-4 in the results and we added a table (Table 5) that summarizes the main clinical features and their frequency in this selected cohort.

Currently, this paper is relevant for a small audience interested in NFIA, but I think even that audience will be confused by some of the things in this paper, which I have pointed out in the attachment (which has some formatting issues, I apologize). I think review of similar types of manuscripts would be useful in finding the best ways to present this data. I also wonder if there are any statistical analyses that could be done (maybe compare to NFIX disorder?)

We greatly appreciate the excellent editing work. We have also encountered formatting issues and we have made the suggested changes in the word file. These changes are highlighted in yellow for keeping track of them.

As far as the statistical suggestion, we think that a careful selection of patients must be performed before any statistical analysis. The careful analysis of patients with NFIX disorder has not been done and this work is beyond the scope of this article. This interesting observation could be a clue for a future paper.

We appreciated and approved all comments made by the Reviewer. However, as said, we have encountered formatting issues, therefore we answer to the comments below.

Commented [GM1]: You have not defined these in the abstract, spell out.

We defined ID/DD and NDD in the abstracts

 Commented [GM2]: indent

Done

 Commented [GM3]: were these not clinical arrays?

Yes, both Agilent and Technogenetics platforms are currently used in the diagnostic workout.

Commented [GM4]: It is assumed that you used only data that passed QC.

Yes, it is. We added “only” in the text.  “…we elaborated only experiments that met the ‘excellent’ criteria as determined by the QC report”

Commented [GM5]: What does this mean?

We further specified the centile of each body parmeter at birth. We decided to include such data in order to give a better description of the clinical case

Commented [GM6]: Brachydactyly?

We replaced “stubby hands” with “brachydactyly”

 Commented [GM7]: Is this relevant? Do other patients have Chiari malformations?

Chiari malformations is a clinical feature often reported in “Chromosome 1p32p31 deletion syndrome”.

 Commented [GM8]: Technically ID is not diagnosed until 5 years of age

We replaced “ID” with “DD”

Commented [GM9]: Again, is this relevant to other

“Skin dyschromia” is a clinical feature reported in “Chromosome 1p32p31 deletion syndrome”.

Commented [GM10]: relevance?

“Neither ventricles anomalies, nor craniosynostosis, were reported”.

Both ventricles anomalies and craniosynostosis are quite characteristic clinical signs of NFIA-haploinsufficiency and their presence or absence have to be evaluated, to calculate their frequency.

Commented [GM11]: I appreciate this! Most of the time you only get the Ensembl or RefSeq

Commented [GM12]: NFIA-haploinsufficiency Genes should be in italics “ex-11” should be “ex 11” DECIPHER should be in all capital letters Dn is not defined in the legend

We have made the changes required.

Commented [GM13]: Label the chr at the top/bottom Label the SNVs

We modified figure 1

Commented [GM14]: Bracydactyly?

We replaced “stubby hands” with “brachydactyly”

Commented [GM15]: Some are lowercase and some uppercase. Also indicate if blank means NO or not evaluated/available.

We modified the lower cases. We added in the blank boxes the sign “-“ and we indicated that it means 'not evaluated'.

Commented [GM16]: This is an eye anomaly, not behavioral.

We moved “mild strabismus” in Table 2 (Dysmorphisms)

Commented [GM17]:Not behavior
We moved “right eye exotropia, mild ptosis”” in Table 2 (Dysmorphisms)

Commented [GM18]: ?

We deleted “hypogenetic”

Commented [GM19]: Possibly physical?

We furtherly expressed such concept by changing it with “facial asymmetry”

Commented [GM20]: Physical?

We changed the sentence with “mild spasticity at lower limbs”

Commented [GM21]: I don’t understand this sentence

“It is worth noting that the comorbidity is an almost constant feature of NDDs and this aspect leaded to suppose that, in most of cases, the different NDDs are just the phenotypic outcome of the same genetic alteration.”

We replaced this sentence with

“It is worth noting that the comorbidity is an almost constant feature of NDDs because different NDDs can represent the variable expressivity of the same genetic alteration.”

Commented [GM22]: This should appear earlier in the manuscript

“Chromosome 1p32-p31 deletion syn-drome”, that is also referred to as "brain malformations with or without urinary tract defects" (BRMUTD) (OMIM # 613735).

We moved this sentence at the beginning of the paper.

Commented [GM23]: Dysgenetic? Dysgenesis?

We replaced “digenetic” with “dysgenetic”

Commented [GM24]: You should add percentages of the total in the tables so we can say X% of individuals with NFIA haploinsufficiency have X trait

We would rather add the percentage in “Table 5” only. The percentages may be not very indicative because some peculiar clinical signs have only been evaluated in a very small number of patients. Phenotypic traits that can facilitate clinicians in identifying the affected patients are not only the more frequents such as ID, but also the more peculiar. For example, the bilateral proximally placed first fingers has been evaluated only in 6 patient and is possible that its frequency is underestimated.

Commented [GM25]: Same as in introduction

In the abstract?

We have modified the sentence in the abstract with

"NFIA-haploinsufficiency can be suspected by a careful observation of the dysmorphisms (macrocephaly, craniofacial and first finger anomalies) and this potential diagnosis is strengthened by the presence of intellectual and developmental disabilities or other neurodevelopmental disorders.  Further clues of NFIA-haploinsufficiency can be provided by instrumental tests such as MRI and kidney urinary tract ultrasound and be confirmed by the genetic testing."

'Further clues of NFIA-haploinsufficiency can be provided by instrumental tests such as MRI and kidney urinary tract ultrasound and be confirmed by the genetic testing'.

Commented [GM26]: I thought that this gene was already thought to be a main contributor.

We replaced “emerges” with “confirms”

Commented [GM27]: This does not need to be in quotes

"Chromosome 1p32p31 deletion syndrome",

We eliminated the quotes.

Commented [GM28]: Needs more of a concluding sentence

“…congenital heart defects, and Juvenile Moyamoya, were never reported in this selected cohort with only NFIA-haploinsufficiency, and probably they are caused by dosage alterations of different deleted genes”.

In this paper the differences between NFIA-haploinsufficiency and Chromosome 1p32p31 deletion syndrome are not thoroughly examined.

The search for genes involved in the clinical signs of Chromosome 1p32p31 deletion syndrome, but not shared by NFIA-haploinsufficiency, is a complex analysis we have started. So far, we have not been able to identify “new disease genes” associated with distinct phenotypical traits of Chromosome 1p32p31 deletion syndrome, but we hope in the future to collect significant data to be presented in a next paper.

You claim to focus on haploinsufficiency, but also have missense variants in your table. Is there evidence that these impair protein function? You could have a table with in silico predictions.
We added in Table 1 the in-silico predictions of the impaired protein function.

You do not need to have as many links in the text as you do, as many of these resources have papers that can be cited.

We agreed and eliminated many links within the text

In Figure 1, a protein level map would be nice to see if they cluster in any domains.

We appreciated the suggestion and modified figure 1, highlighting the functional domains

Reviewer 2 Report

Bertini et al report two individuals with tiny deletions involving NFIA only and undertake a targeted literature review to ascertain individuals who are haploinsufficient for NFIA and no other genes.  This is a useful exercise and well done to highlight the key phenotypic features attributable to NFIA only.  My suggestions for improvement are as follows:

  1. The Introduction should start with the chromosome 1p32p31 region and clinical phenotype first before narrowing down to NFIA and its genetic and biological functions.  It would make more sense that way given this is a very clinically orientated paper. 
  2. Tables 1-4 are rich in detail, but are dense and difficult to extract the main points.  I suggest keeping Table 1 in the main text, but summarising Tables 2-4 by collating the main phenotypes with their overall prevalence e.g. macrocephaly 24/24 (100%).  The full Tables 2-4 can be moved to Supplemental Data instead.  In fact, Tables 2-4 could be summarised into one Table if the authors include the dysmorphism, CNS, neurodevelopmental and urogenital features as subsections of the same Table.  This would make it much easier to digest and understand at a glance.  
  3. There are numerous spelling and grammatical errors throughout.  I suggest the authors use spell check and ask a native English speaker or professional editor to review the manuscript.  These errors reduce the readability of the paper, e.g. “hands and feed”; “dysmorfisms”; “cranialfacial”
  4. There are many blank boxes in each of the tables - what do blank boxes mean?  Not recorded?  The authors then need to make a decision about how to report the prevalence as a fraction when the denominator is variable depending on what each paper has reported.
  5. References are needed for each row in each of Tables 2-4.
  6. I liked the summary of features that the authors have included of the features that they attribute to other genes e.g. cryptorchidism etc.  I think this is worth expanding in the discussion, rather than having it as the final sentence of the Conclusion.  Missed opportunity for a rich discussion about the broader 1p32p31 phenotype
  7.  

Author Response

We thank the Reviewer for his/her work and further updated our manuscript in order to meet his/her expectations.

Bertini et al report two individuals with tiny deletions involving NFIA only and undertake a targeted literature review to ascertain individuals who are haploinsufficient for NFIA and no other genes.  This is a useful exercise and well done to highlight the key phenotypic features attributable to NFIA only.  My suggestions for improvement are as follows:

  1. The Introduction should start with the chromosome 1p32p31 region and clinical phenotype first before narrowing down to NFIA and its genetic and biological functions.  It would make more sense that way given this is a very clinically orientated paper.

We thank for the suggestion and moved the description of Chromosome 1p32p31 deletion syndrome" to the beginning of the introduction.

  1. Tables 1-4 are rich in detail, but are dense and difficult to extract the main points.  I suggest keeping Table 1 in the main text, but summarising Tables 2-4 by collating the main phenotypes with their overall prevalence e.g. macrocephaly 24/24 (100%).  The full Tables 2-4 can be moved to Supplemental Data instead.  In fact, Tables 2-4 could be summarised into one Table if the authors include the dysmorphism, CNS, neurodevelopmental and urogenital features as subsections of the same Table.  This would make it much easier to digest and understand at a glance.

We agreed with the Reviewer and summarized Tables 2-4 in Table 5, which collects the main clinical signs of NFIA-haploinsufficiency and their frequency. We moved tables 2-4 in the results (as suggested by the reviewer 1)

  1. There are numerous spelling and grammatical errors throughout.  I suggest the authors use spell check and ask a native English speaker or professional editor to review the manuscript.  These errors reduce the readability of the paper, e.g. “hands and feed”; “dysmorfisms”; “cranialfacial”

We apologize for the inconvenient, spelling and grammatical errors have been corrected.

  1. There are many blank boxes in each of the tables - what do blank boxes mean?  Not recorded?  The authors then need to make a decision about how to report the prevalence as a fraction when the denominator is variable depending on what each paper has reported.

We added in the blank boxes the sign "-" and we indicated that it means 'not evaluated'.

In Table 5 we specified how prevalence was calculated.

  1. References are needed for each row in each of Tables 2-4.

We added the references in each row in each of Tables 2-4

  1. I liked the summary of features that the authors have included of the features that they attribute to other genes e.g. cryptorchidism etc.  I think this is worth expanding in the discussion, rather than having it as the final sentence of the Conclusion.  Missed opportunity for a rich discussion about the broader 1p32p31 phenotype.

In this paper the differences between NFIA-haploinsufficiency and Chromosome 1p32p31 deletion syndrome are not thoroughly examined.

The search for genes involved in the clinical signs of Chromosome 1p32p31 deletion syndrome, but not shared by NFIA-haploinsufficiency, is a complex analysis we have started. So far, we have not been able to identify “new disease genes” associated with distinct phenotypical traits of Chromosome 1p32p31 deletion syndrome, but we hope in the future to collect significant data to be presented in a next paper.
